# Assessing the Efficacy of Cyanobacterial Strains as *Oryza sativa* Growth Biostimulants in Saline Environments

**DOI:** 10.3390/plants13172504

**Published:** 2024-09-06

**Authors:** Meruyert O. Bauenova, Fariza K. Sarsekeyeva, Asemgul K. Sadvakasova, Bekzhan D. Kossalbayev, Ramazan Mammadov, Aziza I. Token, Huma Balouch, Pavel Pashkovskiy, Yoong Kit Leong, Jo-Shu Chang, Suleyman I. Allakhverdiev

**Affiliations:** 1Faculty of Biology and Biotechnology, Al-Farabi Kazakh National University, Al-Farabi 71, Almaty 050038, Kazakhstan; bauyen.meruyert@gmail.com (M.O.B.); asem182010@gmail.com (A.K.S.); aziza93t@gmail.com (A.I.T.); huma.blch85@gmail.com (H.B.); 2Tianjin Institute of Industrial Biotechnology, Chinese Academy of Sciences, No. 32, West 7th Road, Tianjin Airport Economic Area, Tianjin 300308, China; kossalbayev.bekzhan@gmail.com; 3Ecology Research Institute, Khoja Akhmet Yassawi International Kazakh-Turkish University, Turkistan 161200, Kazakhstan; 4Department of Biology and Ecology, Faculty of Nature and Technology, Odlar Yurdu University, Baku AZ1072, Azerbaijan; rmammad@yahoo.com; 5K.A. Timiryazev Institute of Plant Physiology, Russian Academy of Sciences, Botanicheskaya Street 35, Moscow 127276, Russia; pashkovskiy.pavel@gmail.com; 6Department of Chemical and Materials Engineering, Tunghai University, Taichung 407, Taiwan; yoongkit1014@thu.edu.tw (Y.K.L.); changjs@mail.ncku.edu.tw (J.-S.C.); 7Research Center for Smart Sustainable Circular Economy, Tunghai University, Taichung 407, Taiwan; 8Department of Chemical Engineering, National Cheng Kung University, Tainan 701, Taiwan; 9Department of Chemical Engineering and Materials Science, Yuan Ze University, Chung-Li 32003, Taiwan; 10Institute of Basic Biological Problems, FRC PSCBR RAS, Pushchino 142290, Russia; 11Faculty of Engineering and Natural Sciences, Bahçeşehir University, Istanbul 34353, Turkey

**Keywords:** salinity, cyanobacteria, antioxidant, growth-stimulating activity, sustainable agriculture

## Abstract

Soil salinity, which affects plant photosynthesis mechanisms, significantly limits plant productivity. Soil microorganisms, including cyanobacteria, can synthesize various exometabolites that contribute to plant growth and development in several ways. These microorganisms can increase plant tolerance to salt stress by secreting various phytoprotectants; therefore, it is highly relevant to study soil microorganisms adapted to high salinity and investigate their potential to increase plant resistance to salt stress. This study evaluated the antioxidant activity of four cyanobacterial strains: *Spirulina platensis* Calu-532, *Nostoc* sp. J-14, *Trichormus variabilis* K-31, and *Oscillatoria brevis* SH-12. Among these, *Nostoc* sp. J-14 presented the highest antioxidant activity. Their growth-stimulating effects under saline conditions were also assessed under laboratory conditions. These results indicate that *Nostoc* sp. J-14 and *T. variabilis* K-31 show significant promise in mitigating the harmful effects of salinity on plant size and weight. Both strains notably enhanced the growth of *Oryza sativa* plants under saline conditions, suggesting their potential as biostimulants to improve crop productivity in saline environments. This research underscores the importance of understanding the mechanisms by which cyanobacteria increase plant tolerance to salt stress, paving the way for sustainable agricultural practices in saline areas.

## 1. Introduction

Increasing crop yield, particularly that of cereals, through the use of phototrophic microorganisms, offers a promising solution to food shortages and land degradation. Cyanobacteria, in particular, are advantageous because of their ability to synthesize and release bioactive substances such as auxins, gibberellins, cytokinins, vitamins, polypeptides, and amino acids, which promote plant growth and development [1]. Salt accumulation in irrigated areas is one of the major factors that reduce rice yields, impacting germination, metabolism, plant size, branching, and leaf size [2].

The detrimental effects of salinity on plant growth are linked to pigment degradation and reduced osmotic potential, leading to stomatal closure and decreased CO_2_ fixation [3]. This results in increased photorespiration and the production of reactive oxygen species (ROS) [4]. To counteract the adverse effects of ROS, plants have evolved various antioxidant systems, encompassing nonenzymatic antioxidants and diverse antioxidant enzymes whose levels increase under oxidative stress, including salinity [5]. However, these mechanisms alone may not suffice, as plants under stress demand additional nutrient and energy resources, optimize their hormonal status, and reduce the intensity of salinity stress.

The soils with high salinity and anthropogenic stress conditions harbor unique microflora capable of surviving extreme environmental conditions [6]. The potential of this microflora can be harnessed to alleviate the adverse effects of various stressors on plant growth and development [7]. In this context, microorganisms adapt to thrive under high-salinity conditions, and studies investigating their potential to increase plant resistance to salt are of particular interest. The biodiversity of free-living and associated diazotrophs in saline soils encompasses a wide array of eukaryotic and prokaryotic taxa, among which photosynthetic microorganisms hold particular intrigue [8].

Cyanobacteria and microalgae are among the few microorganisms capable of thriving in desert and saline environments. The most abundant species found in arid saline soils are the *Anabaena*, *Microcoleus*, *Phormidium*, *Nostoc*, *Lyngbya*, and *Aulosira* genera, which are known for their ability to survive and thrive in highly saline environments [9]. Cyanobacteria are particularly notable for their role in enhancing soil fertility, reducing soil pH, improving soil structure, and supporting microbial communities. These microorganisms secrete extracellular polymeric substances, which can bind to sodium ions and form biofilms, thereby protecting plants from salt stress. Cyanobacterial are also capable of removing soluble sodium by biosorption [10]. Additionally, cyanobacteria exhibit a high water retention capacity and increase soil biomass following their death and decomposition. The improved growth of rice, maize, and wheat has been attributed to soil desalinization by cyanobacteria [11,12].

Cyanobacteria, crucial components of soil, have been successfully identified in various medium- and high-salinity soils, including *Nostoc* sp. and *Anabaena* sp., among others [13]. These microorganisms are not only primary nitrogen fixers in agricultural soils but also secrete substances that promote plant growth, such as hormones, vitamins, and polyphenolic compounds, with potent antioxidant activity [14]. Phenolic compounds play a vital role in the antioxidant defense mechanisms of algae and are responsible for their adaptive response to oxidative stress. These natural antioxidants are integral to regulating biochemical and bioenergetic processes in plant cells [15]. Previously, algae and cyanobacteria were believed to lack the enzymes necessary for flavonoid synthesis. However, recent findings using mass spectrometric analysis have confirmed the presence of polyphenols, including flavonoids, in different species of microalgae and cyanobacteria [16]. Several classes of flavonoids are synthesized in these organisms, including isoflavones, flavanones, flavonols, and dihydrochalcones [17]. These studies clearly show that although microalgae and cyanobacteria are primitive organisms compared with higher plants, they can synthesize complex phenolic compounds. The available data allow us to consider the phenolic substances contained in cyanobacteria as components of the antioxidant defense system. Although the total content of polyphenolic compounds in cyanobacteria is insignificant, their amount in biomass increases under intense illumination, exposure to UV radiation, increased temperature, and changes in the composition of the nutrient medium [18]. They play a specific role in the adaptive defense response to oxidative stress.

In addition to phenolic compounds, cyanobacteria also produce growth-promoting metabolites such as auxins, as demonstrated by Shariatmadari et al., 2013 [19] and Hashtroudi et al., 2013 [20], who reported that cyanobacterial extracts increased root length, plant height, and biomass in pumpkin, cucumber, and tomato plants. Extracts of the cyanobacteria *Cylindrospermum muscicola* and *Anabaena oryzae* stimulated the growth of *Lupinus termis* by increasing the chlorophyll content, photosynthetic activity, and nitrogen and carbon in leaves [21]. The extracts contained auxins, gibberellic acid, and cytokinins, which increased the levels of these metabolites in the plants. Different strains of cyanobacteria, such as *Nostoc entophytum* and *Oscillatoria angustissima*, contain different levels of these hormones, resulting in different effects on pea growth [22]. 

Additionally, the growth-stimulating effects of cyanobacteria are associated with the reduction in oxidized substances in the soil, provision of oxygen to the rhizosphere, tolerance to salt stress, and solubilization of phosphates [7,23]. For instance, Guzmán-Murillo [24] successfully alleviated the adverse effects of salt stress on *Capsicum annuum* L. by treating seeds with polysaccharide extracts from the microalgae *Dunaliella salina* and *Phaeodnctylum tricornutum*. Similarly, Mutale-Joan et al., 2021 demonstrated enhanced tolerance of *Solanum lycopersicum* L. plants to salt stress and improved nutrient uptake through the addition of a 5% extract comprising *Dunaliela salina*, *Chlorella ellipsoidea*, *Arthrospira maxima*, and *Aphanothece* sp. [4]. A study tested *Nostoc piscinale* in winter wheat and reported that 0.3 g/L treatment at tillering and ear emergence significantly increased yield by increasing root strength, leaf chlorophyll, and water content [25]. Similar results were obtained using corn plants [26]. 

Despite all the above, it is not entirely clear which specific microalgae strains can be effective in reducing the stress effects on plants and what mechanisms underlie these processes. In this work, we attempted to answer the question of how the accumulation of specific compounds in algae is able to mitigate salinity-induced stress in rice. The objectives of this study were to evaluate the antioxidant profiles and growth-promoting effects of four cyanobacterial strains: *Nostoc* sp. J-14, *T. variabilis* K-31, *O. brevis* SH-12, and *S. platensis* Calu-532 under salinity treatment in the laboratory. We hypothesized that stress-induced phenolic compounds in algae can reduce the toxic effects of salinity.

## 2. Results

### 2.1. Antioxidant Activity of the Examined Cyanobacterial Strains

The antioxidant activity of the studied cyanobacterial strains was determined to identify the most stress-tolerant strains. This activity was measured using an assay involving free synthetic long-lived radicals such as 2,2′-azino-bis-(3-ethylbenzothiazoline-6-sulfonic acid) (ABTS), 2,2-diphenyl-1-picrylhydrazyl (DPPH), and 2,2′-azobis (2-amidinopropane) dihydrochloride (AAPH), with butylated hydroxyanisole (BHA) serving as the standard. 

The results of the antioxidant activity study are shown in Table 1. The *Nostoc* sp. J-14 strain presented the highest antioxidant activity, with an IC50 value of 2.13 ± 0.11 µg/mL, whereas the *T. variabilis* K-31 and *S. platensis* Calu-532 strains showed antioxidant activity values of 2.42 ± 0.22 and 2.81 ± 0.28 µg/mL, respectively. Conversely, the lowest activity was obtained for *O. brevis* SH-12, at 3.21 ± 0.39 µg/mL. In the ABTS assay, *Nostoc* sp. J-14 (IC50 value: 1.58 ± 0.01 µg/mL) and *T. variabilis* K-31 (IC50 value: 1.81 ± 0.01 µg/mL) demonstrated the most promising antioxidant activities. In contrast, *S. platensis* Calu-532 and *O. brevis* SH-12 exhibited relatively low activity (Table 1).

In the ferric reducing antioxidant power (FRAP) experiment, *T. variabilis* K-31 exhibited the highest antioxidant activity, with 2.26 ± 0.16 mg Trolox equivalents (TE)/g extract, followed by *Nostoc* sp. J-14, with 1.33 ± 0.06 mg TE/g extract. In a copper ion-reducing antioxidant capacity assay (CUPRAC), *Nostoc* sp. J-14 showed particularly high antioxidant activity, with a value of 3.32 ± 0.03 mg TE/g extract. In contrast, the other strains, *S. platensis* Calu-532, *T. variabilis* K-31, and *O. brevis* SH-12, had significantly lower activities of 1.30 ± 0.07, 1.22 ± 0.03, and 0.9 ± 0.01 mg TE/g extract, respectively. In the β-carotene/linolic acid analysis, the highest activity was again found for *T. variabilis* K-31 (66.23 ± 1.62%), while the lowest activity was observed for *O. brevis* SH-12 (17.62 ± 1.14%) (Table 1). Each antioxidant assay (DPPH, ABTS, FRAP, CUPRAC, and β-carotene-linoleic acid) complements the others, providing a comprehensive understanding of the antioxidant activity of cyanobacterial extracts. DPPH and ABTS are used to assess the ability to neutralize different radicals, FRAP and CUPRAC measure the reducing potential under different conditions, and β-carotene–linoleic acid assesses membrane protection from oxidation. Using all methods together allows for an objective assessment of antioxidant properties and the identification of synergistic effects of compounds.

Additionally, the quantities of phenolic components present in specific flavonoids of the tested cyanobacterial strains were evaluated. The highest concentrations of phenolic compounds were detected in the *T. variabilis* K-31 and *Nostoc* sp. J-14 strains, with values of 8.08 ± 0.12 and 7.05 ± 0.01 mg, respectively, of quercetin equivalents (QE)/g dry extract (DE). The *O. brevis* SH-12 and *S. platensis* Calu-532 strains contained 6.56 ± 0.13 and 6.29 ± 0.05 mg of QE/g of DE, respectively (Figure 1). The study of the antioxidant properties of the cyanobacterial strains revealed that the tested strains exhibited antioxidant activity, with *Nostoc* sp. J-14 and *T. variabilis* K-31 being the most distinguished strains.

### 2.2. Effects of Different NaCl Concentrations on Cyanobacterial Culture Growth

A study was conducted to examine the functional activity of cyanobacterial cultures under various concentrations of chloride ions. This study was carried out using monocultures of the cyanobacteria *Nostoc* sp. J-14, *T. variabilis* K-31, *O. brevis* SH-12, and *S. platensis* Calu-532. Figure 2 presents the effects of sodium chloride content on the growth and development of cyanobacteria cultures. The cyanobacteria cultures of *Nostoc* sp. J-14 and *T. variabilis* K-31 exhibited a wide growth range across NaCl concentrations ranging from 0 to 16 g/L. In contrast, the *O. brevis* SH-12 and *S. platensis* Calu-532 cultures grew only at NaCl concentrations up to 8 g/L. Among the strains, *Nostoc* sp. J-14 emerged as the most NaCl-tolerant strain, maintaining a significant population of live cells even at high salt concentrations (up to 16 g/L) in the nutrient medium. The critical concentrations of sodium chloride varied across species, ranging from 16 to 24 g/L. The growth rate at these concentrations decreased significantly compared with that at lower concentrations.

Thus, the cyanobacterial strains *Nostoc* sp. J-14 and *T. variabilis* K-31 exhibited the greatest stability among the studied strains. As a result, they were selected for further experimentation to determine their growth-stimulating properties under saline conditions.

### 2.3. Growth-Stimulating Activity of the Tested Strains

The subsequent phase involved examining the growth-stimulating activity of the cyanobacteria *Nostoc* sp. J-14 and *T. variabilis* K-31 in saline soil. Distilled water was used as a control (Figure 3).

The results of the growth-stimulating activity study are presented in Figure 4. The results showed that both tested cyanobacterial strains positively influenced plant growth, as evidenced by the emergence of *O. sativa* shoots in the experimental treatment groups as early as the fourth day. However, as the experiment progressed, a slight divergence in trends was observed. Compared with the control, the plants exposed to salt exhibited reduced shoot growth and root biomass. The cyanobacterial strain *Nostoc* sp. J-14 showed the greatest increase in shoot length at a salt concentration of 2 g/L, with an average shoot length of 25.43 ± 0.05 cm. Notably, there was a slight decrease in the growth parameters of *Nostoc* sp. J-14 plants cultured in media supplemented with 16 g/L salt, resulting in an average shoot length of 22.8 ± 0.08 cm in the experimental group. The cyanobacterial strain *T. variabilis* K-31 was 21.97 ± 0.06 cm long at a concentration of 16 g/L.

The shoot dry weights of the *O. sativa* plants are presented in Figure 5. The highest dry weight of shoot biomass (0.23 ± 0.03 g) was recorded at a salt concentration of 2 g/L when the plants were treated with *Nostoc* sp. J-14 and *T. variabilis* K-31 with a dry weight of 0.21 ± 0.02 g.

## 3. Discussion

The results of the experiment conducted to determine the growth-stimulating effects of cyanobacteria on rice crops in saline soil indicated that *T. variabilis* K-31 and *Nostoc* sp. J-14 exhibited growth-stimulating activity. The investigated cyanobacterial strains *Nostoc* sp. J-14, *T. variabilis* K-31, *O. brevis* SH-12, and *S. platensis* Calu-532 showed increased antioxidant activity and a high content of phenolic compounds. Among these strains, the two most active strains, *Nostoc* sp. J-14 and *T. variabilis* K-31 (Table 1, Figure 1), were selected for further evaluation of their growth-stimulating properties in saline soils.

The effectiveness of these cyanobacteria is probably due to both their production of biologically active substances (Table 1, Figure 1) and their nitrogen-fixing properties. Previous reports indicate that both strains possess high nitrogenase activity [27,28]. Despite the differences between these bacterial species, they appear to employ similar mechanisms to influence plant development. These mechanisms include increasing the availability of nutrient compounds, modulating phytohormonal activity (direct), and reducing the inhibitory effects of various biotic and abiotic factors (indirect). They also act as biopesticides and immunomodulators, further supporting plant growth and resilience [29]. The resistance mechanisms can be activated simultaneously, but at different intensities, which is manifested in the different abilities of the studied algae to reproduce under saline conditions (Figure 2). Additionally, improving soil texture through the growth of phototrophic microorganisms in the plant rhizosphere favours the penetration of salts into deeper soil layers, thereby significantly reducing plant root damage. However, in our experiments, this is unlikely since the plants were grown in pots, where the root system completely filled the volume (Figure 3).

Salinization causes osmotic and toxic effects due to the accumulation of sodium and chlorine ions, which leads to the formation of reactive oxygen species (ROS). As a result, the antioxidant system is activated, including both non-enzymatic and enzymatic antioxidants [30]. We found that the initial flavonoid content in *T. variabilis* was higher than that in *Nostoc* (Figure 1). However, under saline conditions, Nostoc maintained greater biomass growth (Figure 2). We assume that stress-induced substances secreted by *Nostoc* are able to neutralize the effects of salinity to a greater extent, which is confirmed by the greatest length of rice shoots in this variant (Figure 3). Mutale-Joan et al. found that applying 5% microalgae and cyanobacteria extracts improved salt stress tolerance in tomatoes by enhancing vegetative growth, photosynthesis, osmotic regulation, and ion homeostasis, and reducing oxidative stress [4]. Therefore, cyanobacteria may be more effective at enhancing salt tolerance, nutrient uptake, and plant growth, as supported by the findings of Grzesik et al., who reported that foliar treatments with cyanobacteria and algae improved the growth and physiological parameters of willow plants [31]. In our study, we did not investigate hormone levels, but on the basis of the mass gain of rice plants treated with *Nostoc*, it can be assumed that the combined effect of hormones and the intensification of nitrogen accumulation led to a decrease in salinity-induced stress. We can assume that this alga is able to form symbiotic relationships with rice plants [32]. It is not known whether the *Nostoc* strain used in the experiment is capable of synthesizing gibberellins in its exometabolites; however, this phenomenon is widespread among other cyanobacteria [33]. GA is known to neutralize salinity-induced oxidative stress in rice. Although the mechanisms of this process are still poorly understood, it is assumed that gibberellins are involved in this regulation, since the expression of genes associated with the production and inactivation of GA, antioxidants, and α-amylase was detected in rice plants under saline conditions [34].

As a result of these experiments, we discovered a strain of cyanobacteria capable of increasing rice resistance to salinity, but thus far, only under laboratory conditions. We assume that in the future, this may serve as a basis for creating more effective and cost-effective methods of rice cultivation. The results obtained can already be applied in closed greenhouses and farms engaged in rice genetics, where it is especially important to obtain high-quality seeds for further research in the shortest possible time.

It is necessary to study in detail the mechanisms underlying these interactions, but it is already clear that this complex process is caused by both the formation of symbioses and increases in nitrogen fixation and biostimulation due to cyanobacterial exometabolites.

## 4. Materials and Methods

### 4.1. Biological Materials

This study utilized cyanobacterial strains obtained from the collection of phototrophic microorganisms at the Laboratory of Photobiotechnology of Al-Farabi KazNU. These strains were *Nostoc* sp. J-14 (MZ079360), *Trichormus variabilis* K-31 (MZ079356), *Oscillatoria brevis* SH-12 (MZ090011), and *Spirulina platensis* Calu-532, all of which were previously isolated from rice fields in the Republic of Kazakhstan. The collected strains of cyanobacteria were previously characterized as nitrogen-fixing strains and have great potential for use in agricultural biotechnology [27]. The experiments to determine the growth-stimulating activity of the strains were conducted using seeds of the three-line hybrid *Oryza sativa* indica type Nei 5 You (8015 Hybrid rice, Zhejiang Agricultural Science and Technology Seed Industry Co., Ltd., Huzhou, Zhejiang, China).

### 4.2. The Cultivation of and Biomass Production by Microorganisms

The cyanobacterial strains *Nostoc* sp. J-14, *T. variabilis* K-31, *O. brevis* SH-12, and *S. platensis* Calu-532 were grown in a 1 L flask containing 500 mL of BG-11 medium at 27 °C for 7 days under artificial light with a light intensity of 50 μmol (photons) m^−2^ s^−1^. Supplemented air was filtered through a 0.22 µm sterile membrane filter (Millipak, Merck, Darmstadt, Germany). Optical density measurements at 720 nm were taken at desired intervals for 2–3 weeks, depending on the optimum growth rate of each strain, to determine the stationary phase of growth. The cyanobacteria biomass was separated by centrifugation at the stationary phase, and the optical density was adjusted to 1.5 units at 720 nm.

### 4.3. Preparation of Supernatants and Cell Extracts for Determination of Antioxidant Properties

Cyanobacterial cells were harvested during the stationary growth phase by centrifugation using a 5810R centrifuge (Eppendorf, Hamburg, Germany) at 5000 rpm and 23 °C for 15 min. The supernatant was carefully decanted and the precipitated cells were resuspended in 15 mL of pure methanol. This mixture was transferred to a glass extraction bottle and shaken vigorously for 20 min to facilitate the extraction of cellular components. Following the extraction, the mixture was centrifuged again at 5000 rpm for 10 min to separate the methanol extract from the cellular debris. The methanol supernatant was transferred to a clean flask and dried using a rotary evaporator (R-300, Thermo Fisher Scientific, Waltham, MA, USA) under reduced pressure until the solvent was completely evaporated. The dried extracts were then transferred to labelled storage vials and stored at −80 °C in a DW-86L486E freezer (Haier Biomedical, Qingdao, China) for subsequent analyses.

### 4.4. Analysis of Antioxidant Activity

#### 4.4.1. DPPH Free Radical Scavenging Activity Assay

DPPH free radical scavenging activity assays were conducted using five different concentrations of extracts (0.2 mL, 0.4 mL, 0.6 mL, 0.8 mL, and 1.0 mL) and BHA (as a positive control) mixed with 4 mL of a 0.004% methanol solution of DPPH [35]. The mixtures were incubated for 30 min at room temperature in the dark and measured at 517 nm on a spectrophotometer (UV-2600, Shimadzu, Kyoto, Japan). Three replicates of each concentration were measured simultaneously. The DPPH free radical scavenging activity was calculated as follows (Equation (1)):A (%) = (Ac − As)/Ac × 100(1)
where A is the DPPH free radical scavenging activity, Ac is the optical density of the control (methanol), and As is the optical density of the sample, including the extract.

#### 4.4.2. Analysis of ABTS Cation Radical Scavenging Activity

A solution containing ABTS cation radicals was prepared by mixing 7 mM ABTS and 2.45 mM potassium persulfate, which was then diluted with 50% methanol until an initial optical density of approximately 0.70 ± 0.02 at 745 nm was obtained. The solution was subsequently mixed with five different plant extract concentrations (0.05 mg/mL, 0.1 mg/mL, 0.15 mg/mL, 0.2 mg/mL, and 0.25 mg/mL). After incubation for 30 min, the optical density was measured on a spectrophotometer at 734 nm. The results are presented as Trolox equivalents (TUs) relative to the weight of the extract (mg/g).

#### 4.4.3. Analysis of β-Carotene–Linoleic Acid

The antioxidant activity of the cyanobacterial extracts was evaluated using the β-carotene–linoleic acid test system with slight modifications [36]. Initially, a solution of 0.2 mg of β-carotene in 1 mL of chloroform was prepared, followed by the addition of 20 µL of linoleic acid and 200 mg of a Tween-20 (Sigma, Saint Louis, MO, USA) emulsifier mixture. The mixture was then evaporated at 40 °C for 10 min using a rotary evaporator to remove the chloroform. After evaporating the chloroform, 4.8 mL of the emulsion was added to test tubes containing 0.2 mg of the sample and 0.2 units of the extract. For the control, 0.2 mL of the solvent (methanol, ethanol, or acetone) was added to the test tubes instead of the extract. Once the emulsion was added to the test tubes, the initial optical density at 470 nm was measured using a Shimadzu UV-1601 spectrophotometer (Kyoto, Japan). The measurements were performed at 30 min intervals for 2 h. All samples were analysed in triplicate, with butylated hydroxytoluene used as a standard. The antioxidant activity was quantified by measuring the extent of β-carotene bleaching, which was calculated as follows (Equation (2)):AA = [1 − (A0 − At/A0_o_ − At_o_) × 100](2)
where AA is the total antioxidant activity, A0 is the sample’s initial absorbance, At is the control’s initial absorbance, A0_o_ is the sample’s absorbance after 120 min, and At_o_ is the control’s absorbance after 120 min.

#### 4.4.4. Copper Ion-Reducing Antioxidant Capacity Assay

The copper ion-reducing activity of the extracts was investigated according to the method of Apak et al. [37]. A volume of 0.5 mL of extract solution was mixed with a premixed mixture containing CuCl_2_·2H_2_O (1 mL, 10^−2^ M), neocuproine (1 mL, 7.5 × 10^−3^ M), and ammonium acetate buffer (1 mL, 1 M, pH 7.0). Following a 30 min incubation period, the absorbance at 450 nm was recorded on a UV-2600 spectrophotometer (Shimadzu, Kyoto, Japan). The results are presented as Trolox equivalents (TUs) relative to the weight of the extract (mg/g).

#### 4.4.5. Ferric Reducing Antioxidant Power Assay

The FRAP assay was conducted following the methodology outlined by Benzie and Strain [38]. One milliliter of extract solution was combined with 2 mL of FRAP reagent, which was prepared by mixing a 300 mM sodium acetate buffer solution at pH 3.6, 10 mM TPZT (2,4,6-tris(2-pyridyl)-s-triazine) in 40 mM HCl, and 20 mM FeCl_3_ at a ratio of 10:1:1 (*v*/*v*/*v*). The tubes were incubated for 30 min, after which the optical density was measured at 593 nm on a UV-2600 spectrophotometer (Shimadzu, Japan). The results are presented as Trolox equivalents (mg TU/g).

#### 4.4.6. Analysis of Total Flavonoid Content

The total flavonoid content of the extracts was determined as quercetin equivalents (mg QE/g) using the aluminum colorimetric method [39]. For each extract, 1 mL of methanol solution (100 μg/mL) was mixed with 1 mL of aluminum trichloride (AlCl_3_) in methanol (2%). The absorbance was measured on a spectrophotometer (UV-2600, Shimadzu, Kyoto, Japan) at 415 nm after 10 min against a blank sample of 1 mL of methanol and 1 mL of plant extract without AlCl_3_. The total flavonoid content was determined using a standard curve, with quercetin as the standard. The average of three readings expressed in mg of quercetin equivalents (QE) per 100 mg of extract or fraction was used.

### 4.5. Effect of Different NaCl Concentrations on the Survival Rate of Cyanobacteria

A survival test was conducted to determine the lethal dose of NaCl for each strain. The strains were cultivated under laboratory luminostat conditions with an illumination intensity of 50 μmol (photons) m^−2^ s^−1^ on the medium surface. BG-11 nutrient media containing 0, 2, 4, 8, 16, or 24 g/L NaCl (or 0, 34.2, 68.4, 136.8, 273.6, and 410.4 mM NaCl, respectively) were prepared. Each experimental culture was inoculated with 6 × 10^5^ cells from medium-logarithmic exponential growth cultures of the cyanobacterial strains. Three replicates of each NaCl concentration were prepared. The biomass accumulation of the cultures was determined by measuring the optical density of the ethanol extracts using a spectrophotometer (1021, CECIL, Cambridge, UK). The experiment was conducted over a duration of seven days.

### 4.6. Determining the Growth-Stimulating Properties of Selected Cyanobacterial Strains in Saline Soil

Soil samples with a pH of 7.69, organic matter content of 17.80 g/kg, total nitrogen content of 3.00 g/kg, available nitrogen content of 37.33 mg/kg, total phosphorus content of 0.39 g/kg, available phosphorus content of 9.57 mg/kg, total potassium content of 8.87 g/kg, and available potassium content of 61.84 mg/kg were obtained from a rice field in the Almaty region of Kazakhstan. The soil samples were air-dried, thoroughly mixed, and sieved using a 0.5 cm sieve to remove plant residues. Thirteen *O. sativa* seeds were sown into each plastic pot (8 cm diameter, 10 cm height) containing 240 g of artificially salinated soil. The pots were divided into different salinity levels (0, 2, 4, 8, and 16 g/L or 0, 34.2, 68.4, 136.8, and 273.6 mM NaCl, respectively) to test the salinity tolerance of the cyanobacterial isolates (*n* = 3). After 5 days, 11 *O. sativa* plants whose growths were similar were retained. The soil in the pots was then moistened with 30 mL of the prepared cyanobacterial liquid inoculum (1 × 10^8^ cells mL^−1^) or an equivalent volume of water.

Three treatments were carried out as follows:(1)Soil soaked with *T. variabilis* K-31;(2)Soil soaked with *Nostoc* sp. J-14;(3)Soil soaked with water.

The experiment spanned 16 days, comprising three repetitions of each treatment. Pots were housed in a growth chamber (CM 4/50-295 RR, SM Climate, Saint Petersburg, Russia) maintained at a relative humidity of 75%. Prior to seedling emergence, the temperature was held constant at 30 °C. Lighting was provided by fluorescent lamps delivering 100 μmol (photons) m^−2^ s^−1^ for 16 day/8 dark, measured using a Mastech MS6612 digital light meter (Mastech, Moon Twp, PA, USA). Subsequently, the temperatures were set to 28 °C during the day and 24 °C at night for the single-leaf stage, 28 °C during the day and 25 °C at night for the two-leaf stage, and 28 °C during the day and 22 °C at night for subsequent stages. Pots received 30 mL of water every 48 h, and their positions were randomized periodically.

Six repetitions of each treatment were randomly selected 16 days after sowing for further analysis. The plants from each pot were harvested and carefully separated into roots and shoots to determine growth parameters, including length, fresh weight, and dry weight. A ruler was used to measure the lengths, while an electronic scale (Sartorius, Göttingen, Germany) was used to determine the fresh and dry weights. According to the results obtained during the experiment, the average length and the standard error of the arithmetic mean of the seedlings were calculated using Excel.

### 4.7. Statistical Analysis

The experiments included 5 biological replicates and 3 analytical replicates. The significance of the differences among the experimental groups was calculated by one-way analysis of variance (ANOVA) followed by Duncan’s method using SigmaPlot 12.3 (Systat Software Inc., Chicago, IL, USA). The letters above the columns indicate significant differences among the different options. The data are shown as the mean ± SD.

## 5. Conclusions

We demonstrated that the application of cyanobacterial strains significantly enhanced the growth of *O. sativa* plants under laboratory conditions. The high antioxidant activity of cyanobacterial strains, specifically *T. variabilis* K-31 and *Nostoc* sp. J-14, indicates their potential role in improving plant tolerance and growth efficiency under saline conditions. Research into the use of cyanobacterial strains as growth biostimulants in saline environments is crucial, but it represents only an initial step toward a comprehensive understanding of their mechanisms. Investigating the pathways through which cyanobacteria produce antioxidants, including enzymes, genes, and activation conditions under salinity stress is essential. To understand how cyanobacteria improve plant tolerance to salt stress, it is important to study their effects on ion transporters, osmolyte accumulation, and signalling pathways. It is important to study the mechanisms of symbiosis formation with rice plant roots and to investigate the processes of nitrogen fixation. These findings contribute to our understanding of the complex interactions within the soil microbiome.

## Figures and Tables

**Figure 1 plants-13-02504-f001:**
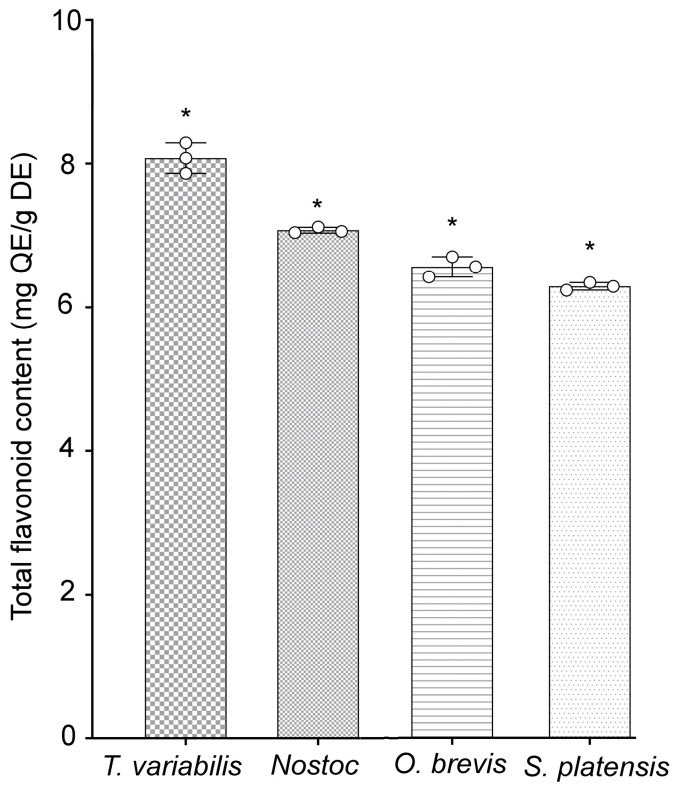
Total flavonoid contents of cyanobacterial extracts. Different asterisks indicate the levels of statistical significance: * *p* < 0.05, *n* = 5. QE = quercetin equivalents; DE = dried extract.

**Figure 2 plants-13-02504-f002:**
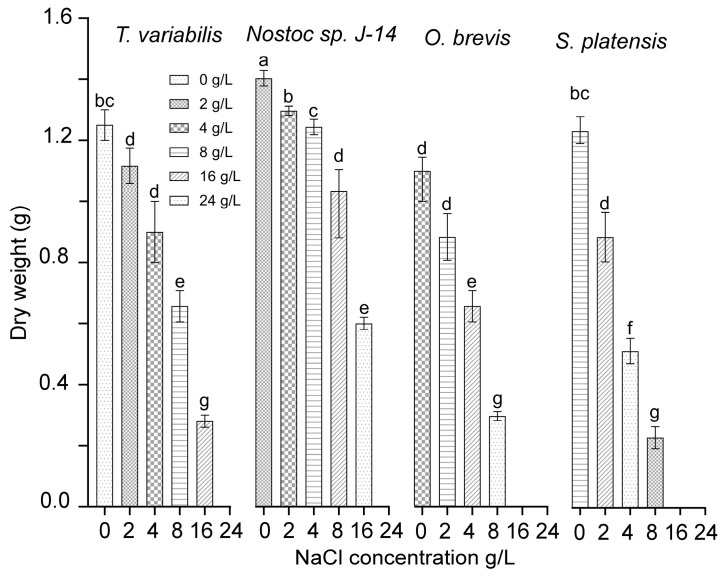
Dry biomass cyanobacterial cultures at different NaCl concentrations. The mean values ± SDs are shown. Different letters indicate significant differences at *p* < 0.05, *n* = 5.

**Figure 3 plants-13-02504-f003:**
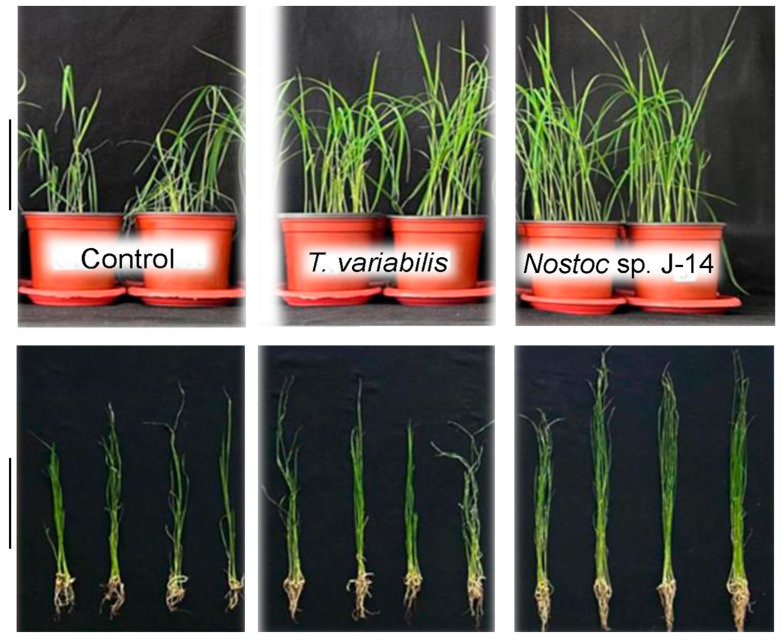
The impact of the combined action of cyanobacteria utilized in the experiment and NaCl treatment on the growth of 16-day-old *O. sativa* plants. The scale bar in the figure is 10 cm.

**Figure 4 plants-13-02504-f004:**
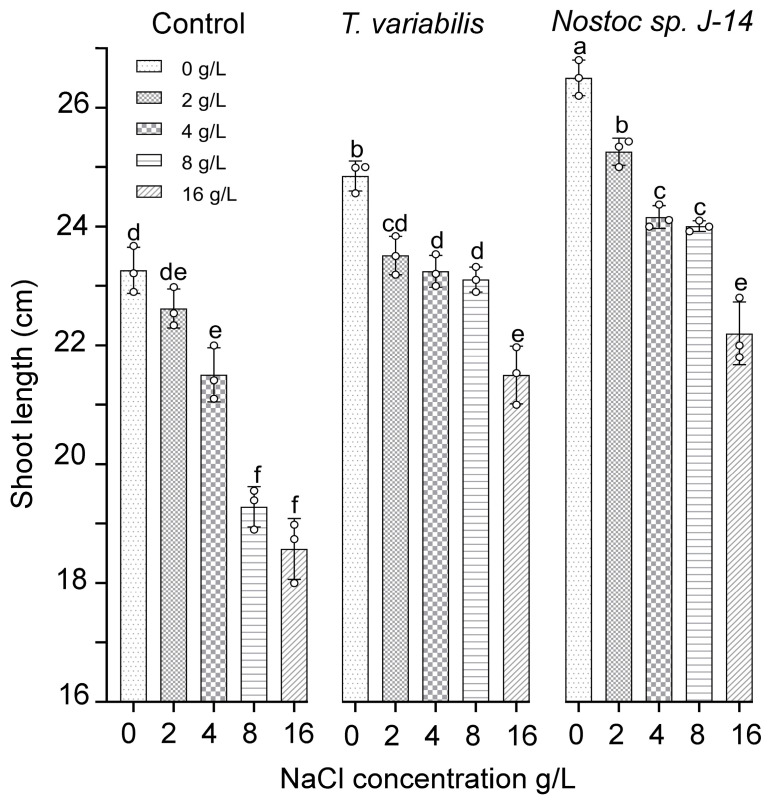
The impact of the combination of cyanobacteria and NaCl on the shoot length of 16-day-old *O. sativa* plants. The mean values ± SDs are shown. Different letters indicate significant differences at *p* < 0.05, *n* = 5.

**Figure 5 plants-13-02504-f005:**
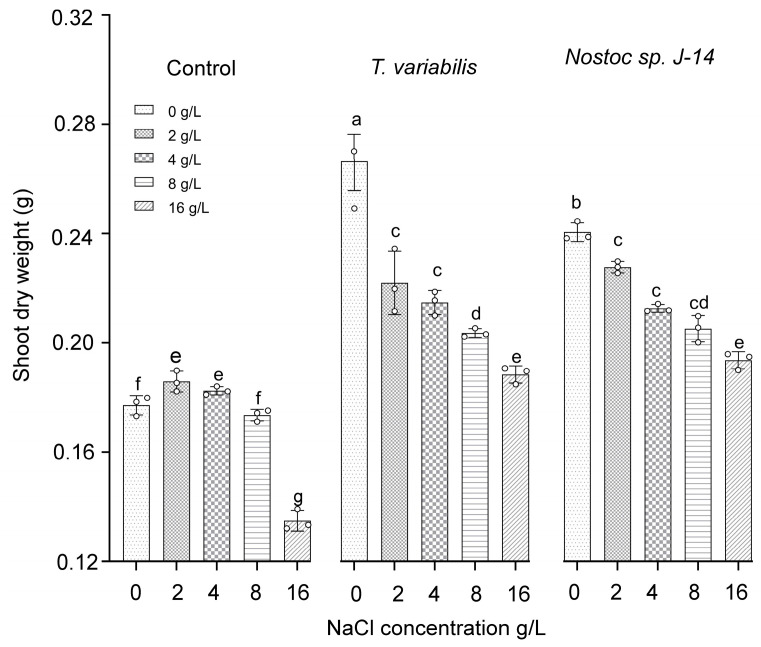
The impact of the combination of cyanobacteria and NaCl on the shoot dry weight of 16-day-old *O. sativa* plants. Different letters indicate significant differences at *p* < 0.05, *n* = 5.

**Table 1 plants-13-02504-t001:** Antioxidant activity of the algae extracts.

Extracts	DPPH ^1^	ABTS ^2^	FRAP ^3^	CUPRAC ^4^	β-Carotene/ Linoleic Acid Assay
	The IC50 Value (μg/mL)	mg TE/g Extract			(%)
*T. variabilis* K-31	2.42 ± 0.22 c	1.81 ± 0.21 a	2.26 ± 0.16 a	1.22 ± 0.03 b	66.23 ± 1.62 a
*Nostoc* sp. J-14	2.13 ± 0.11 d	1.58 ± 0.12 b	1.33 ± 0.06 b	3.32 ± 0.03 a	20.66 ± 3.73 c
*O. brevis* SH-12	3.21 ± 0.39 a	0.72 ± 0.01 c	0.50 ± 0.01 d	0.90 ± 0.01 d	17.62 ± 1.14 d
*S. platensis* Calu-532	2.81 ± 0.28 b	0.62 ± 0.02 d	0.26 ± 0.01 c	1.30 ± 0.07c	38.75 ± 0.88 b

^1^ 2,2-diphenyl-1-picrylhydrazyl ferric reducing antioxidant power assay; ^2^ 2,2′-azino-bis (3-ethylbenzothiazolin-6-sulfonic acid); ^3^ ferric reducing ability of plasma; ^4^ copper ion-reducing antioxidant capacity assay. The mean values ± SDs are shown. Different letters indicate significant differences at *p* < 0.05, *n* = 5.

## Data Availability

Data are contained within the article.

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
