# Peer review of "Assessing the Efficacy of Cyanobacterial Strains as *Oryza sativa* Growth Biostimulants in Saline Environments"

_plants, 2024, doi:10.3390/plants13172504_

Round 1
Reviewer 1 Report
Comments and Suggestions for Authors
Dear Authors, I have reviewed the manuscript, and my comments are given below.
The theme of the manuscript is that Nostoc sp. A J-14 and T. variabilis K-31 show significant promise in mitigating the adverse effects of salinity on plant size and weight. Both strains significantly enhanced the growth of Oryza sativa plants under saline conditions, suggesting their potential as biostimulants to improve yield in saline environments.
The title and abstract are appropriate, as are the keywords.
Introduction: suggest rewriting the chapter. This chapter is very superficial, with 8 literature references in total. It does not present the results of the topic in a comprehensive way and no parallel can be drawn between the Introduction and Discussion chapters. The Discussion chapter contains a large number of references that are included here for the first time. It should not be written in this way. They should appear first in the Introduction chapter and the Discussion chapter should be used to compare them with your own results. In addition, it is worth drawing on results published in the last 5 years.
Author Response
- Introduction: suggest rewriting the chapter. This chapter is very superficial, with 8 literature references in total. It does not present the results of the topic in a comprehensive way and no parallel can be drawn between the Introduction and Discussion chapters. The Discussion chapter contains a large number of references that are included here for the first time. It should not be written in this way. They should appear first in the Introduction chapter and the Discussion chapter should be used to compare them with your own results. In addition, it is worth drawing on results published in the last 5 years.
Answer: This has been done. We completely rewrote the Abstract and Discussion sections.
Reviewer 2 Report
Comments and Suggestions for Authors
High salinity is a significant environmental stress factor that hampers plant growth by degrading pigments and reducing osmotic potential, leading to stomatal closure, decreased CO2 fixation, and increased production of reactive oxygen species (ROS). Plants have developed antioxidant systems to combat ROS, but these alone are insufficient under stress conditions. Soils with high salinity host unique microflora that can help alleviate plant stress. Cyanobacteria, such as Nostoc sp. and Anabaena sp., are crucial in saline soils, acting as nitrogen fixers and secreting growth-promoting substances. These microorganisms improve soil structure and enhance plant tolerance to salt stress by reducing oxidized substances, providing oxygen, and solubilizing phosphates. Studies have shown that treating plants with extracts from these microalgae can significantly enhance their resistance to salt stress and improve nutrient uptake. This paper is about the evaluation of the antioxidant profile and growth-stimulating effect of some cyanobacterial strains. My comments, questions:
Line 72-86: more references are needed in the introduction, with exact results about the effect of cyanobacterial strains or extracts on plants.
Line 91: please provide some information about why these antioxidant activities (DPPH, ABTS, FRAP, CUPRAC, β-carotene-linoleic acid assay) and the flavonoids were evaluated.
Line 186: most of the paragraphs contain data derived from the experiment, I miss the comparison of these data with some other experiments referred in the References chapter.
Line 208-244: I suggest inserting these paragraphs into the Introduction chapter, as they contain introductory information related to the parameters studied.
Line 258-278: it is the same as what I mentioned previously.
Line 288-322: I cannot find any data from the experiment, these paragraphs only for provide references without any comparison of the experimental results obtained.
Line 327-328: you state, “application of specific cyanobacterial strains can positively impact soil health and plant nutrition”, there is not any data about soil microorganisms or nutrients, I do not recommend using this sentence.
Line 344-353: what was the cultivation time of cyanobacterial strains? 7 days or 23- weeks?
Line 492: please use the expression “growth biostimulants” as you did not study any plant hormone in this experiment.
Later, I suggest you conduct more experiments in field condition to justify the laboratory results.
Author Response
- Line 72-86: more references are needed in the introduction, with exact results about the effect of cyanobacterial strains or extracts on plants.
Answer: This has been done. We improved the paragraph.
Cyanobacteria, a crucial component of soil, have been successfully identified in various medium- and high-salinity soils, including Nostoc sp. and Anabaena sp., among others [4]. These microorganisms not only are primary nitrogen fixers in agricultural soils but also secrete substances that promote plant growth, such as hormones, vitamins, and polyphenolic compounds, with potent antioxidant activity [5]. Cyanobacteria produce growth-promoting metabolites such as auxins, as demonstrated by Shariatmadari et al. [Shariatmadari Z., Riahi H., Seyed Hashtroudi M., Ghassempour A., Aghashariatmadary Z. Plant growth promoting cyanobacteria and their distribution in terrestrial habitats of Iran. Soil Sci. Plant Nutr. 2013;59:535–547. doi: 10.1080/00380768.2013.782253.] and Hashtroudi et al. [Hashtroudi M.S., Ghassempour A., Riahi H., Shariatmadari Z., Khanjir M. Endogenous auxins in plant growth-promoting Cyanobacteria—Anabaena vaginicola and Nostoc calcicola. J. Appl. Phycol. 2013;25:379–386. doi: 10.1007/s10811-012-9872-7. ], who reported that cyanobacterial extracts increased root length, plant height, and biomass in pumpkin, cucumber, and tomato plants. Extracts of the cyanobacteria Cylindrospermum muscicola and Anabaena oryzae stimulated the growth of Lupinus termis by increasing the chlorophyll content, photosynthetic activity, and nitrogen and carbon contents in leaves [Haroun S.A., Hussein M.H. The promotional effect of algal biofertilizers on the growth, protein patterns and metabolic activities of Lupinus termis plants growing in siliceous soil. Asian J. Plant Sci. 2003;2:944–951. doi: 10.3923/ajps.2003.944.951.]. The extracts contained auxins, gibberellic acid and cytokinins, which increased the levels of these metabolites in the plants. Different strains of cyanobacteria, such as Nostoc entophytum and Oscillatoria angustissima, contain different levels of these hormones, resulting in different effects on pea growth [Osman M.E.H., El-Sheekh M.M., El-Naggar A.H., Gheda S.F. Effects of two species of cyanobacteria as biofertilizers on metabolic activities, growth, and yield of pea plants. Biol. Fertil. Soils. 2010;46:861–875. doi: 10.1007/s00374-010-0491-7.]. Their capacity to retain moisture, facilitated by their jelly like cell structure, enhances the soil structure by loosening it [6]. Additionally, the growth-stimulating effects of cyanobacteria are associated with the reduction of oxidized substances in the soil, provision of oxygen to the rhizosphere, tolerance to salt stress, and solubilization of phosphates [7,8]. For example, Guzmán-Murillo [9] successfully alleviated the adverse effects of salt stress on Capsicum annuum L. by treating seeds with polysaccharide extracts from the microalgae Dunaliella salina and Phaeodnctylum tricornutum. Similarly, Mutale-Joan et al. demonstrated enhanced tolerance of Solanum lycopersicum L. plants to salt stress and improved nutrient uptake through the addition of a 5% extract comprising Dunaliela salina, Chlorella ellipsoidea, Arthrospira maxima, and Aphanothece sp. [1]. A study tested Nostoc piscinale in winter wheat and reported that 0.3 g/L treatment at tillering and ear emergence significantly increased yield by increasing the root strength, leaf chlorophyll content, and water content [Takács, G., Stirk, W. A., Gergely, I., Molnár, Z., van Staden, J., & Ördög, V. (2019). Biostimulating effects of the cyanobacterium Nostoc piscinale on winter wheat in field experiments. South African Journal of Botany, 126, 99-106.]. Similar results were obtained for corn plants [Ördög, V., Stirk, W. A., Takács, G., Pőthe, P., Illés, A., Bojtor, C., ... & Nagy, J. (2021). Plant biostimulating effects of the cyanobacterium Nostoc piscinale on maize (Zea mays L.) in field experiments. South African Journal of Botany, 140, 153-160. ].
- Line 91: please provide some information about why these antioxidant activities (DPPH, ABTS, FRAP, CUPRAC, β-carotene-linoleic acid assay) and the flavonoids were evaluated.
Answer: Each antioxidant assay (DPPH, ABTS, FRAP, CUPRAC, and β-carotene-linoleic acid) complements the others, providing a comprehensive understanding of the antioxidant activity of cyanobacterial extracts. DPPH and ABTS are used to assess the ability to neutralize different radicals, FRAP and CUPRAC are used to measure the reducing potential under different conditions, and β-carotene-linoleic acid is used to assess membrane protection from oxidation. Using all methods together allows for an objective assessment of antioxidant properties and the identification of synergistic effects of compounds.
We added this information to the text.
- Line 186: most of the paragraphs contain data derived from the experiment, I miss the comparison of these data with some other experiments referred in the References chapter.
Answer:
It is done. We completely rewrote the Abstract and discussion sections.
- Line 208-244: I suggest inserting these paragraphs into the Introduction chapter, as they contain introductory information related to the parameters studied.
Answer: This has been done.
- Line 258-278: it is the same as what I mentioned previously.
Answer: This has been done.
- Line 288-322: I cannot find any data from the experiment, these paragraphs only for provide references without any comparison of the experimental results obtained.
Answer:
It is done. We completely rewrote the discussion section.
- Line 327-328: you state, “application of specific cyanobacterial strains can positively impact soil health and plant nutrition”, there is not any data about soil microorganisms or nutrients, I do not recommend using this sentence.
Answer: We agree and cross out this senescence.
- Line 344-353: what was the cultivation time of cyanobacterial strains? 7 days or 2-3 weeks?
Answer: First, the algae were transferred to culture containers and allowed to adapt for 1 week. The strains were subsequently cultivated for an additional 2‒3 weeks depending on the strain and when the optical density reached the required level.
- Line 492: please use the expression “growth biostimulants” as you did not study any plant hormone in this experiment.
Answer: We agree. It is done.
- Later, I suggest you conduct more experiments in field condition to justify the laboratory results.
Answer: We are grateful to the Reviewer for this comment. Future research will focus on field studies.
Reviewer 3 Report
Comments and Suggestions for Authors
The Authors should clearly state in their paper that this is a study carried out in laboratory and in pot experiments. Specific mention to this very relevant aspect should be made at lines 41, 89, and 489 (Conclusion) by adding the sentence "at laboratory level and in pot experiments".
Comments on the Quality of English Languageminor editing needed
Author Response
- The Authors should clearly state in their paper that this is a study carried out in laboratory and in pot experiments. Specifically, mention to this very relevant aspect should be made at lines 41, 89, and 489 (Conclusion) by adding the sentence "at laboratory level and in pot experiments".
Answer: We agree. It is done.
- Comments on the Quality of English Language
minor editing needed
Answer: We agree. It is done.
Round 2
Reviewer 1 Report
Comments and Suggestions for Authors
That is good, thank you - I recommend it for publication.
Reviewer 3 Report
Comments and Suggestions for Authors
the comments by reviewer have been considered, the discussion has been entirely re-written. The ms is now acceptable